# Unique Huygens-Fresnel electromagnetic transportation of chiral Dirac wavelet in topological photonic crystal

**Xing-Xiang Wang** [1,2,4], **Zhiwei Guo** [3,4], **Juan Song**[3], **Haitao Jiang**[3], **Hong Chen** [3] ✉ & **Xiao Hu** [1,2] ✉

Light propagates in various ways depending on environment, including uniform medium, surface/interface and photonic crystals, which appears ubiquitously in daily life and has been exploited for advanced optics technology. We unveiled that a topological photonic crystal exhibits unique electromagnetic (EM) transport properties originating from the Dirac frequency dispersion and multicomponent spinor eigenmodes. Measuring precisely local Poynting vectors in microstrips of honeycomb structure where optics topology emerges upon a band gap opening in the Dirac dispersion and a *p-d* band inversion induced by a Kekulé-type distortion respecting $C_{6v}$ symmetry, we showed that a chiral wavelet induces a global EM transportation circulating in the direction counter to the source, which is intimately related to the topological band gap specified by a negative Dirac mass. This brand-new Huygens-Fresnel phenomenon can be considered as the counterpart of negative refraction of EM plane waves associated with upwardly convex dispersions of photonic crystals, and our present finding is expected to open a new window for photonic innovations.

According to the Huygens–Fresnel principle, light propagates in such a way that every point on a wavefront is the source of wavelet, and these secondary wavelets emitted from different points interfere with each other and form the next wavefront[1,2]. This gives an intuitive and unified picture for reflection, diffraction and refraction of light depending on the environment, which appears ubiquitously in daily life and has been exploited for various optics technologies[3]. In photonic crystals (PhCs), which are media with spatially varying dielectric permittivity and/or magnetic permeability, electromagnetic (EM) waves are described by periodic wavefunctions in unit cells and additional Bloch phases[4–6]. The frequency dispersion relation defined in the reciprocal momentum space works as a new notch for light guiding. As a matter of fact, negative refraction generated by the upwardly convex dispersion with a negative band mass in the vicinity of lower band edge[7–9] and zero

refractive index emerging from the Dirac dispersion[10] have been achieved and exploited for modern photonic applications.

Transforming the concept of band topology fostered in electron systems[11–15] to EM waves as initiated by Haldane and Raghu[16,17] opens a completely new platform for harnessing light propagation, where the underlying bosonic feature of the light becomes important[18–22]. It has been demonstrated that EM waves propagate unidirectionally along the interface between a topological and a topologically trivial optic structures with the frequency set inside the common frequency band gap. Due to the topological protection, topological interface EM transports are robust against sharp corners, disorder and randomness inevitable in fabrication process, thus exhibiting huge potential for high-density photonic integrated circuits yearned for advanced photonic technology. As the explicit photonics example of topological

[1]Research Center for Materials Nanoarchitectonics (MANA), National Institute for Materials Science (NIMS), Tsukuba 305-0044, Japan. [2]Graduate School of Science and Technology, University of Tsukuba, Tsukuba 305-8571, Japan. [3]MOE Key Laboratory of Advanced Micro-Structured Materials, School of Physics Science and Engineering, Tongji University, Shanghai 200092, China. [4]These authors contributed equally: Xing-Xiang Wang, Zhiwei Guo. ✉e-mail: hongchen@tongji.edu.cn; Hu.Xiao@nims.go.jp

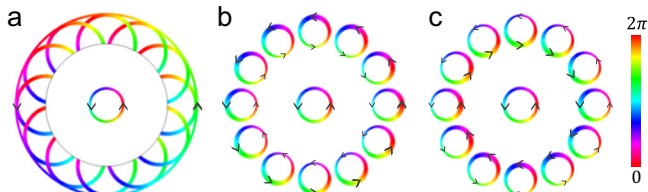

**Fig. 1 | Schematics of Huygens–Fresnel principle for chiral wavelet. a–c** are for continuum dielectric medium, trivial and topological photonic crystals, respectively. Color represents the phase of electromagnetic wave, and arrows indicate the transportation direction of electromagnetic energy. In **b** and **c** the thickness of circles and arrows corresponds the intensity of wavefunction and energy flow. In all cases, the wave propagation is outward, and the length scale in **a** should be much larger than the size of the chiral source, whereas in **b** and **c** it is the unit cell of photonic crystal. On the scale much larger than the size of unit cell, the behavior of the trivial photonic crystal can be described by **a** while that of the topological photonic crystal is completely different.

crystalline insulator[23], a two-dimensional (2D) honeycomb PhC based on conventional semiconductor has been explored in recent years, where the photonic topology emerges upon a gap opening in the Dirac frequency dispersion and a *p-d* band inversion induced by a Kekulé-type distortion respecting $C_{6v}$ symmetry[21]. It is shown that a chiral source with frequency in the band gap can induce unidirectional interface EM transportations, which provides a very useful handle for piloting topological photonic transports uneasy to achieve in other platforms[24–27]. As a matter of fact, it is revealed recently that exciton-polariton and phonon-polariton transports can be guided with the help of the topological EM mode, which is manipulated by an incident chiral light beam from free space[28,29], and that the topological inter-facial modes sustain stable lasing superior to conventional whispering-gallery-modes[30–32]. Far-field probing from the topological metasurface and a high-performance surface-emitting laser based on a new light confinement mechanism due to the inverted *p* and *d* band configurations were also reported[33,34]. These stimulating findings trigger new curiosity on the bulk transportation of chiral Dirac EM modes in the passband with frequency close to band edges.

Here, we show that, as a unique manifestation of Huygens–Fresnel principle, a chiral wavelet (see Fig. 1a) induces EM waves whirling in the same direction as the source (see Fig. 1b, c) in individual $C_{6v}$-symmetric unit cells of the PhC. Remarkably in the topological PhC interference between the *p*- and *d*-modes in these unit cells results in a global EM transportation circulating around the source in the opposite direction to the chiral source (see Fig. 1c). We perform a proof-of-concept experiment in microstrips, a high-frequency waveguide formed by patterned metallic wires on the top of dielectrics/metal substate which is commonly used in microwave devices. The Huygens–Fresnel EM transportation in the honeycomb topological microstrip formed by judiciously tuned thickness of top metallic wires is successfully demonstrated by the circulating Poynting vectors, in complete agreement of theory and full-wave computer simulations.

## Results
### Theory
The **k·p** theory of the Kekulé-distorted honeycomb PhC with $C_{6v}$ symmetry is described by the following 2D massive Dirac Hamiltonian (see Supplementary Note I)

$$H = \begin{bmatrix} -M & -i\nu k_+ & 0 & 0 \\ i\nu k_- & M & 0 & 0 \\ 0 & 0 & -M & -i\nu k_- \\ 0 & 0 & i\nu k_+ & M \end{bmatrix} \quad (1)$$

where the basis is taken as the representations of $E_1$ and $E_2$ of the point group $C_{6v}$: $(|p_+\rangle|d_+\rangle|p_-\rangle|d_-\rangle)$ with eigen orbital angular momentum (OAM) $l = +1\hbar, +2\hbar, -1\hbar, -2\hbar$ respectively, and $-M$ is the squared frequency of *p*-mode seen from the band gap center. The linear off-diagonal entrees in the Dirac Hamiltonian (1)

$$k_\pm = k_x \pm i k_y = (k_R \pm i k_\theta)e^{\pm i\theta} \quad (2)$$

govern the interference between the two modes $|p_+\rangle$ and $|d_+\rangle$ (as well as $|p_-\rangle$ and $|d_-\rangle$), where the momentum is measured from Γ point of the Brillouin zone (BZ) for the real-space $C_{6v}$-symmetric unit cell (see Supplementary Note II); $\nu > 0$ is the relativistic velocity. The Dirac mass $M$ yields a gap with sign in the spectrum of the PhC, especially a negative Dirac mass $M < 0$ is associated with a *p-d* band inversion in frequency, which generates the nontrivial photonics topology[21,35]. Hamiltonian (1) is block diagonalized into pseudospin + and − subspaces, which cannot be mixed in the vicinity of Γ point where only linear terms $k_\pm$ are involved. This property is crucial for EM transportation in the PhC when the frequency is close to the band edges. It is worth noticing that in the present system the pseudospin in the Dirac Hamiltonian (1) is intimately related to the OAM which specifies whirling real-space energy flow in $C_{6v}$-symmetric unit cells (see Fig. 2a, b), in a stark contrast to the spin-block diagonalized BHZ model for the prominent quantum spin Hall effect (QSHE)[36].

To unveil the physics associated with this unique feature, we consider, without losing generality, the pseudospin + subspace where the two modes $|p_+\rangle$ and $|d_+\rangle$ are mixed away from Γ point as

$$|P_+\rangle = \frac{1}{w}\left(|p_+\rangle - \frac{i\nu k_-}{2M}|d_+\rangle\right), |D_+\rangle = \frac{1}{w}\left(|d_+\rangle - \frac{i\nu k_+}{2M}|p_+\rangle\right) \quad (3)$$

upon a perturbation treatment on the off-diagonal entrees of the Dirac Hamiltonian (1) with $w = \sqrt{1 + \left(\frac{\nu k}{2M}\right)^2}$ and $k^2 = k_+ k_-$. As indicated explicitly in the two-component spinor wavefunction Eq. (3), interference between the two modes $|p_+\rangle$ and $|d_+\rangle$ with opposite parity depends on the sign of Dirac mass in an intriguing way. Its physical consequence can be best appreciated when one watches the wave spreading in the PhC induced by a chiral wavelet (for comparison with a uniform dielectric medium see Fig. 1a). For this purpose, we consider a chiral wavelet with counterclockwise phase winding at the system center as displayed in Fig. 2a (see also Fig. 1b, c), which induces wavefunctions spanned by $|p_+\rangle$ and $|d_+\rangle$ in all other $C_{6v}$-symmetric unit cells of the PhC due to conservation of total angular momentum for frequency close to the band edge at Γ point irrespective of the Dirac mass sign.

Noticing that in Eq. (3) $k_-$ adds a phase $\exp(-i\theta)$ to $|d_+\rangle$ (see Fig. 2a, b) in the $C_{6v}$-symmetric unit cell at the azimuthal angle $\theta$, $|p_+\rangle = \exp(i\phi)$ and $|d_+\rangle = \exp(2i\phi)$ enhance (suppress) each other at $\phi = \theta$ ($\phi = \theta + \pi$), *i.e.* the outer (inner) side of the unit cell seen from the system center in the topologically trivial PhC with $M > 0$, as depicted in Fig. 2c (see Supplementary Note II). Therefore, globally a net EM transportation circulates counterclockwise around the system center, in the same direction of the chiral source. In a Huygens–Fresnel picture, the unit cells with whirling energy flow behave as the secondary chiral source, and the azimuthal angle $\theta$ dependence of the relevant momentum $k_-$ (and $k_+$ likewise) glues the real and reciprocal spaces in waveguiding, as a clear manifestation of the Dirac frequency dispersion in Eq. (1) and the two-component spinor wavefunction in Eq. (3), which is defined in the real-space $C_{6v}$-symmetric unit cell distinct from the BHZ Hamiltonian for QSHE. On large scale, the trivial photonic crystal behaves in the same way as a continuum dielectric medium (see Fig. 1a, b).

In a stark contrast, in the topological PhC with $M < 0$ $|p_+\rangle$ and $|d_+\rangle$ enhance (suppress) each other at the inner (outer) side of individual unit cells according to Eq. (3), as displayed in Fig. 2d. Remarkably, this results in a net EM transportation circulating clockwise around to the

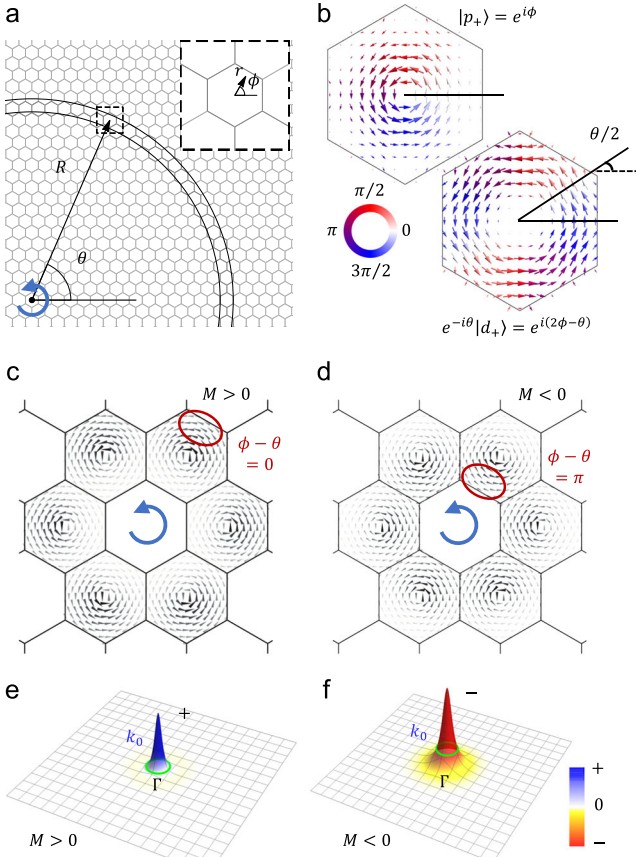

**Fig. 2 | System with honeycomb structure where a chiral source locates at the center. a** A counterclockwise-phase-winding chiral source is adopted, and the response at point $r = (r, \phi)$ inside the unit cell with center at $R = (R, \theta)$ is investigated. **b** Schematic for the interference between the two modes $|p_+\rangle$ and $|d_+\rangle$ in a unit cell at the azimuthal angle $\theta$ for $|P_+\rangle$ as given in the first part of Eq. (3). A similar picture is available for $|D_+\rangle$. **c, d** Schematics for local Poynting vector enhanced at $\phi - \theta = 0$ and $\phi - \theta = \pi$, which generate counterclockwise and clockwise global EM energy flow around the system center for $M > 0$ and $M < 0$, in the same and opposite direction to the source, respectively. **e, f** Berry phase $\gamma_n$ enclosed by the loop of $k_0(\omega)$ surrounding $\Gamma$ point for $M > 0$ and $M < 0$, respectively.

system center, in the opposite direction of the chiral source. Therefore, while in the Huygens–Fresnel picture the secondary wavelet in the topological PhC is similar to the trivial PhC, the global EM transportations are opposite in the two cases (see Fig. 1). It is straightforward to see that the same discussions are available for $|D_+\rangle$ mode given in Eq. (3), and the case of clockwise chiral source is available by considering the time-reversal symmetry.

In order to enhance the above discussions based on the lowest-order perturbation, we perform the second-order perturbation analysis (see Supplementary Note III), where terms mixing the two pseudospin sectors are included additionally into the 4 × 4 Hamiltonian (1) (see Eq. (A24)). It is shown that the wavefunction excited by a chiral source with counterclockwise phase winding in the unit cell coincides with the pseudospin-up eigenstate given by Eq. (3) in the lowest order of $k$, whereas modifications only appear as higher-order infinitesimals $O(k^3)$ which can be neglected safely as far as the physics around band edges is concerned. This verifies the picture based on the first-order perturbation theory.

## Experimental realization

We design a proof-of-concept experiment for the above physics based on microstrips. In the microwave photonics regime, the 2D microstrip loaded with lumped circuit elements is a convenient and simple platform for realizing arbitrary 2D lattices and observing optical wave propagations in a precise way[37,38]. So far, many high-performance microstrip structures have been constructed to achieve various novel optical responses and enable advanced applications, such as superlens[39], near-field routing[40], hyperbolic dispersion[41,42], and topological photonics[43]. In this platform, one benefits two merits: on one hand the microstrip can be modeled by a textbook LC circuit which facilitates a comprehensive theoretical analysis on the advanced Dirac physics; on the other hand, a chiral source matching to the whole structure can be realized based on the $C_{6v}$-symmetric unit cell and, thanking to its completely flat structure, both the amplitude and phase of the out-of-plane electric field can be measured precisely in experiments, from which the local Poynting vector sketching explicitly the EM transportation can be mapped out experimentally.

We consider a honeycomb microstrip and its lumped-element LC circuit shown in Fig. 3a, b, where every site is connected to each of its three neighbors by an inductor and meanwhile is grounded by a capacitor[43]. For uniform inductance $L$ and capacitance $C$, one obtains linear frequency-momentum dispersions with much the same physics of graphene (see Fig. 3f)[44–47]. Making inductance inside the hexagonal unit cells ($L_1$) different from that between unit cells ($L_2$) as shown in Fig. 3c, d, Kekulé distortions respecting $C_{6v}$ symmetry open a band gap in the otherwise double Dirac cones at $\Gamma$ point of BZ as seen in Fig. 3e, g. It is straightforward to check that the k·p theory of this LC circuit is given by Dirac Hamiltonian (1) in units $\omega_0^2 = 1/(L_1 C)$ with $M = (1 - \tau)\omega_0^2$, $v = \omega_0^2 \tau a_0/2$ and $\tau = L_1/L_2$ in terms of the lumped inductors and capacitor and the lattice constant $a_0$, where the center of band gap is at $(2 + \tau)\omega_0^2$ (see Supplementary Note I). Experimentally, the metallic wires inside the $C_{6v}$-symmetric unit cells are tuned thicker (thinner) than those between the unit cells to achieve $M > 0$ ($M < 0$) in Hamiltonian (1) (See also Supplementary Note IV). We put a microwave source with phase winding at the center of a sufficiently large system with a given Dirac mass $M$, and clad the central part by a dual system with opposite Dirac mass, which levitates the target state to reduce undesired environment influences[34]. The chiral source is realized experimentally by connecting six delay lines with the equal phase difference to the six sites of the central $C_{6v}$-symmetric unit cell (see Methods and Supplementary Note V).

In microstrips the TM mode with finite $E_z$, $H_x$ and $H_y$ ($E_x$, $E_y$, $H_z = 0$) is realized, where the in-plane magnetic field is given by the out-of-plane electric field $E_z$ from Faraday's law of induction $\mathbf{H} = -\frac{i}{\mu_0 \omega} \nabla \times \mathbf{E}$. The time-averaged Poynting vector is evaluated by $\langle \mathbf{S} \rangle = \text{Re}\{\mathbf{E}^* \times \mathbf{H}\}/2$, which can be measured experimentally with high precision in the microstrip system, and maps explicitly both local and global EM transportation.

The distributions of the strength and phase of the out-of-plane electric field $E_z$ measured in experiments (see Methods and Supplementary Note V for details of the experiment setup) are depicted in Fig. 4a, b for the sample shown in Fig. 3c for the trivial structure with $M > 0$, which is stimulated by a $+2\pi$-phase-winding chiral wavelet located at the center $C_{6v}$-symmetric unit cell (see also Supplementary Note V, VI) with frequency slightly below the lower band edge. In all individual unit cells marked by the dashed lines the phase winds counterclockwise, in the same way as the source, which clearly evidences that the wavefunctions are of up pseudospin spanned by $|p_+\rangle$ and $|d_+\rangle$, namely half of whole basis, as discussed in the above theory. From the precise amplitude and phase of the out-of-plane electric field $E_z$, the local Poynting vector can be mapped out in the present microstrip system where a TM mode is realized (see Supplementary Note I, II). As displayed in Fig. 4c, the local Poynting vector whirls in individual unit cells counterclockwise same to the source as a direct consequence of the phase winding in Fig. 4b. It is clear that, as read from Fig. 4d, the Poynting vectors summed in individual unit cells form a counterclockwise EM energy flow with respect to the system center,

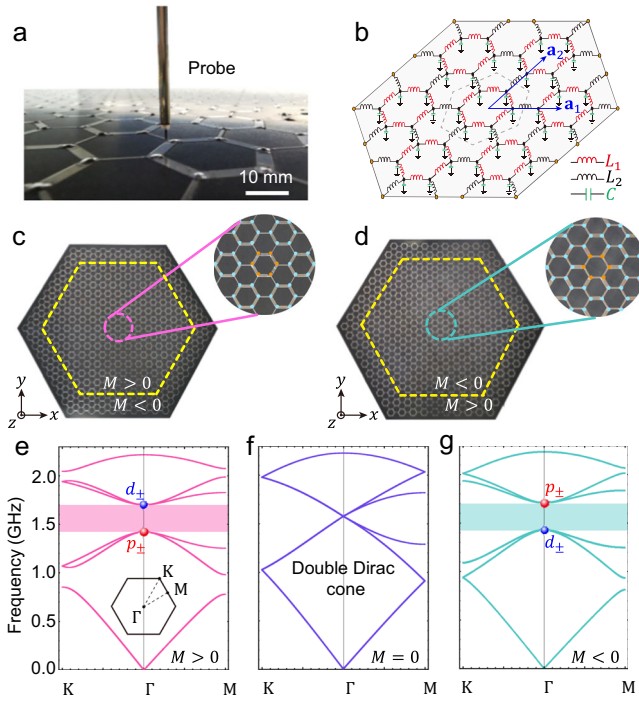

**Fig. 3 | Realization of honeycomb microstrip-based LC-circuits. a** Photo of the experimental setup with an electric-field probe placed right above the microstrip array, which is used to measure distributions of the amplitude and phase of the out-of-plane electric field $E_z$. The metallic strips intra/inter $C_{6v}$-symmetric unit cell have widths of $w_1$ and $w_2$, and the corresponding distributed inductance are $L_1$ and $L_2$, respectively, inversely proportional to the width. A lumped capacitor of 5.6 pF is loaded on the nodes. The corresponding on-node capacitance used for calculations of the frequency band structures is taken as $C = 7.27$ pF, which is slightly larger than the lumped one due to the distributed capacitances coming from the microstrip lines. **b** Effective LC-circuit model based on the honeycomb microstrip with lumped elements. The $C_{6v}$-symmetric unit cell is marked by the dashed lines and two unit vectors are indicated by blue arrows. **c** Photo of the honeycomb microstrip-based structure with $M > 0$ ($w_1 = 2.6$ mm, $L_1 = 3.6$ nH and $w_2 = 0.9$ mm, $L_2 = 6.59$ nH) wrapped by the structure with $M < 0$ ($w_1 = 1.5$ mm, $L_1 = 5.09$ nH and $w_2 = 3.2$ mm, $L_2 = 3.13$ nH), with the interface marked by the dashed yellow lines. The upper inset shows the zoomed-in view of the sample around the center, where the chiral source is marked by orange dots. The lower inset shows the coordinate. **d** Same as **c** except for that the honeycomb microstrip-based structure with $M < 0$ is cladded by the structure with $M > 0$. **e–g** Bulk band dispersions of the LC circuits with $M > 0$, $M = 0$ and $M < 0$ calculated along high-symmetry directions of the Brillouin zone shown in the inset of **e**. The band gaps painted with pink and cyan for $M > 0$ and $M < 0$ are associated with the $p$-$d$ band inversion.

indicating that the trivial microstrip with $M > 0$ responds to the chiral source in a "paramagnetic" way as discussed theoretically. The corresponding results obtained by full-wave simulations are displayed in Fig. 4e–h for comparison, which agree completely with the experimental results.

We carry out measurements on the topological structure with $M < 0$ shown in Fig. 3d where a $+4\pi$-phase-winding chiral source with frequency slightly below the lower band edge is used which matches the eigenmode given in Fig. 3g (see also Supplementary Note VI). It is clear that the experimental results in Fig. 5a–c and the full-wave simulation results in Fig. 5e–g look similar to the corresponding ones in Fig. 4, especially the phase winding and the whirling direction of the local Poynting vector in individual $C_{6v}$-symmetric unit cells are counterclockwise same as the chiral source. In stark contrast, however, as seen in Fig. 5d, h, the Poynting vectors summed in individual unit cells form a clockwise EM transportation, which is in opposition

to the chiral source. Therefore, our experiments demonstrate unambiguously that the topological microstrip with $M < 0$ responds to the chiral source in a "diamagnetic" way, in full agreement with the theoretical analysis. This is the experimental demonstration of the unique Huygens–Fresnel EM transportation in the topological photonics platform.

We have confirmed that when the phase winding of the source is reversed the local Poynting vectors are reversed, which is required by the time-reversal symmetry of the present system, whereas the diamagnetic and paramagnetic responses remain unchanged for $M < 0$ and $M > 0$ respectively, which is determined by the sign of Dirac mass.

## Discussions

### Huygens–Fresnel picture

In Huygens–Fresnel picture people understand the destination of EM wave in terms of phase interferences among stimulated, secondary sources. For continuum dielectric media, wavefronts and the secondary wavelets can be chosen arbitrarily. For the spreading of chiral wavelet in the present honeycomb PhCs, the secondary wavelet is to be taken as the $C_{6v}$-symmetric unit cell. As observed in our experiments and full-wave computer simulations, a chiral source located in the center unit cell induces whirling EM flows in neighboring unit cells in the same direction, which is determined by the phase winding and specifies the pseudospin state; in turn these unit cells induce the same chiral states in outer unit cells, and so on so forth. Namely all $C_{6v}$-symmetric unit cells can be considered as secondary chiral sources, a situation rendered uniquely by the Dirac Hamiltonian Eq. (1). Moreover, the Dirac mass $M$ (including its sign) and the Bloch wavenumber in the chiral form $k_\pm$ govern the phase interferences between the chiral $p$- and $d$-eigenmodes inside individual unit cells in such a special way as given in Eq. (3), which then forms the global EM transportation around the chiral source in opposite directions for $M > 0$ and $M < 0$. Therefore, the peculiar spreading of chiral wavelet in the topological photonics structure unveiled in the present work can be understood as a hybridization of the real-space Huygens–Fresnel principle and the reciprocal-space Dirac frequency dispersion, which are glued by the momentum-governed interference between the eigenmodes with opposite parity hosted in real-space $C_{6v}$-symmetric unit cells in a way specified by the negative Dirac mass. These new features of the chiral response observed in the photonic Dirac systems add new facet to the Huygens–Fresnel picture.

### Berry-phase origin

It is found (see Supplementary Note VII) that the OAM with respect to the system center contributed from a shell of unit cells at $R \gg a_0$ (see Fig. 2a) $\mathbf{L}_g/\hbar = 2\mu_0\omega\sum_{|\mathbf{R}|=R}\mathbf{R} \times \iint_{\text{u.c.}}\langle\mathbf{S}\rangle d\mathbf{r}$ is dominated by the Berry phase $\gamma_n$ enclosed by the loop of $k_0(\omega)$ surrounding $\Gamma$ point

$$\begin{aligned}\mathbf{L}_g/\hbar &= \sum_{|\mathbf{R}|=R}\mathbf{R} \times \iint_{\text{u.c.}}\text{Im}\left(E_z^*\boldsymbol{\nabla}E_z\right)d\mathbf{r} = \mathbf{e}_z\frac{\alpha}{4\pi c^2}[f(R)]^2\oint_{|\mathbf{k}|=k_0}d\mathbf{k}\cdot\mathscr{A}_n(\mathbf{k})\\&= \mathbf{e}_z\frac{\alpha}{4\pi c^2}[f(R)]^2\gamma_n\end{aligned}$$

$$(4)$$

where $\alpha = \tau(2-\tau)a_0^2\omega_0^2/(4|1-\tau|) > 0$ is the coefficient of the quadratic term of the dispersion relation near $\Gamma$ point $\omega_\mathbf{k}^2 = \omega_{\text{edge}}^2 - \alpha k^2/2$ with $\omega_{\text{edge}}^2 = (2 + \tau - |1-\tau|)\omega_0^2$, $[f(R)]^2 = \iint_{\text{u.c.}}\epsilon E_z^* E_z d\mathbf{r}$ (note $-2\hbar\omega_{\text{edge}}/\alpha$ gives the negative band mass in the vicinity of the lower band edge[8], see Supplementary Notes II, VII). Because $\gamma_n > 0$ for $M > 0$ (see Fig. 2e and Supplementary Note VIII), the EM energy flow around the system center is counterclockwise, and the global OAM is positive in the same direction as the source, thus a paramagnetic response. In contrary, for $M < 0$, $\gamma_n < 0$ as shown in Fig. 2f which yields a negative global OAM, and the EM energy flow around the system center is clockwise, in opposition to the source. Therefore, the system with negative Dirac

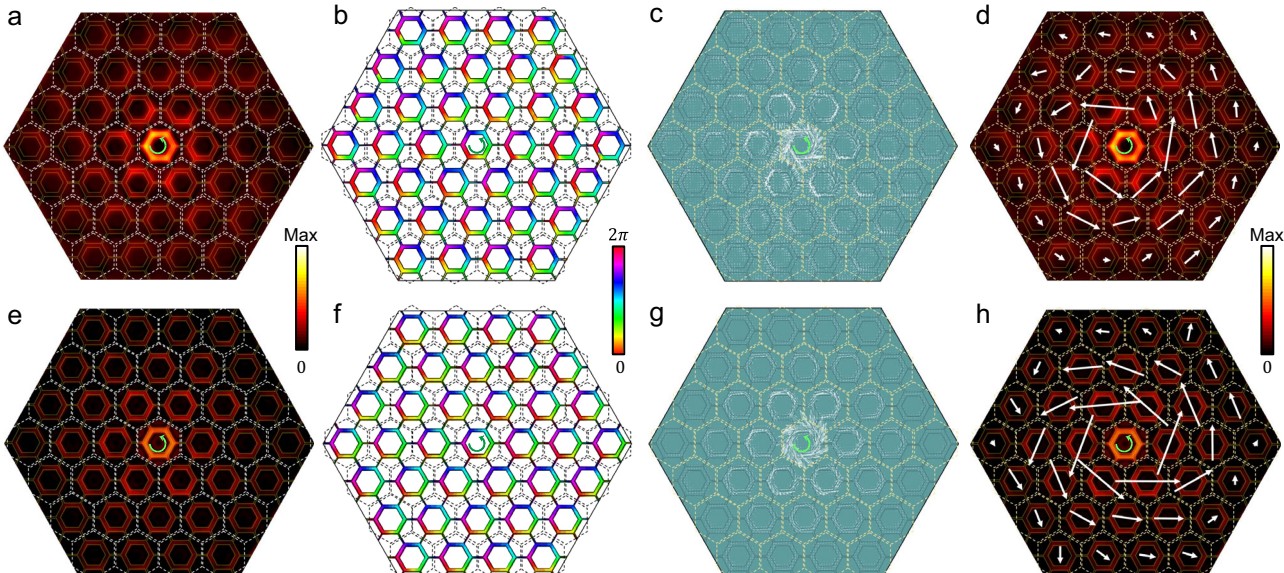

**Fig. 4 | Response of the structure with $M > 0$ to a chiral source with counterclockwise phase winding. a** Distribution of the strength of out-of-plane electric-field $E_z$ in the central part of the sample, which is obtained by the experimental measurements using a $+2\pi$-phase-winding chiral source located at the sample center at $f = 1.43$ GHz. $C_{6v}$-symmetric unit cells are marked by the dashed white lines as the guide for eye. **b** Distribution of the phase of the out-of-plane electric-field $E_z$ under the same condition as **a**. In all $C_{6v}$-symmetric unit cells (marked by the dashed black lines) the phase winds $+2\pi$ same as the source. **c** Distribution of local Poynting vectors obtained by the amplitude **a** and phase **b** of the out-of-plane electric-field $E_z$. Local Poynting vectors rotate counterclockwise in individual unit cells (marked by dashed yellow lines), in the same direction of the source. **d** Distribution of Poynting vectors summed in individual $C_{6v}$-symmetric unit cells obtained from **c**. The Poynting vectors summed in individual unit cells rotate counterclockwise with respect to the sample center, same to the source, corresponding to the para-magnetic chiral response. **e**–**h** Same as **a**–**d**, but for the results obtained by the full-wave simulations at $f = 1.42$ GHz.

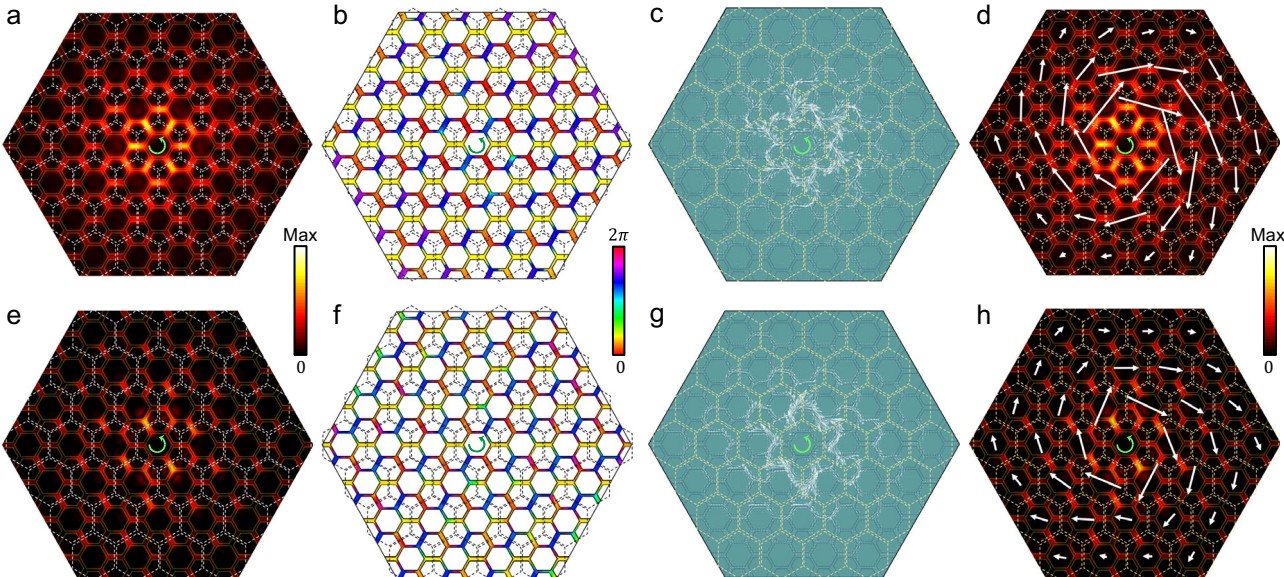

**Fig. 5 | Response of the structure with $M < 0$ to a chiral source with counterclockwise phase winding. a** Distribution of the strength of out-of-plane electric-field $E_z$ in the central part of the sample, which is obtained by the experimental measurements using a $+4\pi$-phase-winding source located at the center $C_{6v}$-symmetric unit cell of the sample at $f = 1.445$ GHz. **b** Distribution of the phase of the out-of-plane electric-field $E_z$ under the same condition as **a**. In all $C_{6v}$-symmetric unit cells the phase winds $+4\pi$ same as the source. **c** Distribution of local Poynting vectors obtained from the amplitude (**a**) and phase (**b**) of the out-of-plane electric-field $E_z$. Local Poynting vectors in individual unit cells rotate counterclockwise, in the same direction of the source. **d** Distribution of Poynting vectors summed in individual $C_{6v}$-symmetric unit cells obtained from **c**. The Poynting vectors summed in individual unit cells rotate clockwise with respect to the sample center, opposite to the source, corresponding to the diamagnetic chiral response. **e**–**h** Same as **a**–**d**, but for the results obtained by the full-wave simulations at $f = 1.42$ GHz.

mass exhibits a diamagnetic response to the chiral source, in agreement with the above theoretical discussions on the interference between the chiral *p*- and *d*- modes.

## Comparison with topological interfacial transportation

The bulk chiral response explored in the present work takes place in passbands with frequency close to the band edges, which is distinct crucially from the topological interfacial transportation for frequency

falling in the bulk band gap. Firstly, in order to realize the topological interfacial modes one has to put the topological and trivial photonic structures back to back in real space, since the Jackiw-Rebbi mechanism requires two Dirac masses with opposite signs[48], which is not the case of the unique chiral response in passband induced by the negative Dirac mass. In the present work we clad the one of them by the other (see Fig. 3) in order to reduce possible noises, which is purely a technical reason and not a must in principle. As a matter of fact, full-wave computer simulations on large enough pure structures, either topological or trivial, exposed to air (namely without cladding dual microstrip) show clearly that the features demonstrated in Figs. 4, 5 remain unchanged (see Supplementary Note X).

Secondly, the real-space positioning of chiral source also leaves different impacts on these two phenomena. For frequency in passband, the pseudospin of EM mode specified by the direction of phase winding or equivalently the whirling EM flow in individual $C_{6v}$-symmetric unit cells is always same to the source, no matter whether the chiral source is located in the center $C_{6v}$-symmetric unit cell (see Figs. 4, 5) or in the $C_{3v}$-symmetric area among three unit cells (see Supplementary Note XI). Therefore, placing the chiral source in the $C_{6v}$-symmetric unit cell and/or in the $C_{3v}$-symmetric area among three unit cells does not change the bulk chiral response (which itself depends on the sign of Dirac mass). In contrast, for frequency in band gap, the whirling direction of local Poynting vector in $C_{6v}$-symmetric unit cells, namely the pseudospin of the EM modes, is the same as the chiral source only when the chiral source is located in a $C_{6v}$-symmetric unit cell, while it is opposite to the source if the chiral source is located in the $C_{3v}$-symmetric area among three unit cells (see Supplementary Note XI). Therefore, placing the chiral source in the $C_{6v}$-symmetric unit cell and/or in the $C_{3v}$-symmetric area among three unit cells results in opposite topological interfacial transportations[49].

Thirdly, while topological interfacial modes are intimately related to the topological invariant index derived by integrating Berry curvatures over the whole BZ manifesting the bulk-edge correspondence of band topology, the chiral response in passband under concern is governed by the Berry curvature in a small region around Γ point in the reciprocal momentum space (see Supplementary Note VII). In addition, the chiral source breaks the time-reversal symmetry by picking up one of the two pseudospin subspaces. Therefore, whether the topological invariant index for the present honeycomb photonics structure is $Z_2$ or not[35,50] is irrelevant to the chiral response under concern.

Since the chiral responses associated with the pseudospin and global OAM fostered in the honeycomb structure highlighted in the present work are bulk optical features, they are compatible to a broader scope of optical applications, including sensor, filter, as well as tweezer and spanner for particle manipulation[34,51–57]. Superior to defect-type structures[51,58], the periodic lattice structure in the present approach can be developed to generate multi-singularity optical vortices useful for high-resolution far-field microscopy[59].

## Negative refraction
The possibility of the negative refraction has been recognized for half century[7], where at the interface between air (for example) and a medium with negative refractive index the directions of incident and refracted light beams locate on the same side of the normal of interface. Later it has been established that the negative refractive index can be realized in PhCs with the frequency below the band gap at Γ point[8,9], where the parallel component of momentum of an EM plane wave keeps the same direction, as the conservation law of momentum, whereas the direction of EM energy transportation is opposite to the incident beam due to the negative band mass associated with the upwardly convex frequency dispersion. Independent of the sign of band gap, negative refraction appears in both trivial and topological photonic structures (see Supplementary Note XII).

For the chiral Dirac modes explored in the present work, OAM in individual $C_{6v}$-symmetric unit cells (namely pseudospins) are parallel to the chiral source, as the conservation law of angular momentum, whereas the global EM energy transportation is in the direction opposite to the source in topological photonic structure with negative Dirac mass $M < 0$ below and above the band gap as can be read from Eqs. (1) and (3).

Both realized in the vicinity of band edges in photonics structures but due to different mechanisms, the peculiar spreading of chiral wavelet in the topological photonics structure unveiled in the present work can be taken as the counterpart of negative refraction of EM plane waves. The combination of the negative refraction and unique chiral response can provide an unprecedented platform for manipulating EM propagation which should be useful for advanced optic applications.

In a general viewpoint, band mass plays an important role in the Bloch scheme for quantum and classical waves in periodic structures, such as negative refraction in photonic systems and positive carrier charge of quasiparticle (namely hole) in electronic crystals featured by negative band mass. As showcased in the present work, Dirac mass associated with the band gap opened in a Dirac dispersion upon real-space construction can be exploited as a unique notch for manipulating wavefunction, and especially negative Dirac mass is expected to trigger intriguing new physics.

## Methods
### Numerical simulations
All full-wave simulations finite-element are performed using Computer Simulation Technology Microwave Studio software based on a finite integration method in the time domain. The structure is constructed on a commercially printed F4B circuit board, which has a relative permittivity and thickness of $\varepsilon_r = 2.2$ and $h = 1.6$ mm, respectively. The loss tangent of the F4B substrate is 0.0079. The thickness and the conductivity of the metallic microstrip lines are set to be 18 μm and $5.8 \times 10^7$ S/m, respectively. As an input port we apply a voltage of 1 V between a node on the top honeycomb structure and the corresponding point on the bottom conductor of the microstrip line platform, and construct a chiral source by decreasing the phase of voltage clockwise (counterclockwise) between neighbor nodes in a metallic hexagon. The impedance of discrete port is 50 Ω, which is consistent with the experimental environment. Open boundary conditions are applied to all three-dimensional directions. The internal resistance in each lumped capacitor is taken as 1 Ω.

### Experimental setup and details of measurement
For the experimental process, the chiral source is designed by an antenna array, which is composed of six antennas with different phase delays. Specially, the signals are generated from the port 1 of a vector network analyzer (Agilent PNA Network Analyzer N5222A), and then divided into six channels through a 1-6 microwave power divider. Each channel output from the power divider is matched with a delay line to control the phase. Thus, the positively-phase-winding chiral source can be realized by increasing ΔΦ counterclockwise between neighbor antennas. Similarly, the negatively winding chiral source can be realized by increasing ΔΦ clockwise between neighbor antennas. The phase difference in the system of $M > 0$ and $M < 0$ are taken as $\Delta\Phi = \pi/3$ and $\Delta\Phi = 2\pi/3$, respectively, to match the eigenmodes below the lower band edge. At last, the six antennas constituting the chiral source are connected one-to-one to the six end points of the unit cell at the center of the sample.

The metal used to design the microstrip is copper, and tin is plated on the surface of the metal to avoid oxidation. Especially, the copper in the microwave regime can be seen as the perfect electric conductor, and the loss of copper can be ignored. The sample is put on a 10-cm-thick foam substrate with a permittivity of near one and then placed on an automatic translation device with scanning steps of 1 mm,

which makes it feasible to accurately to probe the field distribution using a near-field scanning measurement. An electric probe (small rod antenna) of 5 mm length connecting to the port 2 of analyzer is vertically placed 1 mm above the samples to measure the signals of out-of-plane electric field $E_z$. By analyzing the recorded field values, we obtain the distributions of the amplitude and phase of the out-of-plane electric field $E_z$.

## Data availability
All the data that support the findings of this study are available from the corresponding authors upon reasonable request, following the policy of JST.

## Code availability
All the codes that support the findings of this study are available from the corresponding authors upon reasonable request, following the policy of JST.

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

## Acknowledgements

This work is partially supported by the National Key Research Program of China (No. 2021YFA1400602), the National Natural Science Foundation of China (Grant Nos. 12004284, and No. 61621001), the Fundamental Research Funds for the Central Universities (Grant No. 22120210579), and the Shanghai Chenguang Plan (Grant No.21CGA22) (H.C. and Z.G.), and partially by CREST, JST (Core Research for Evolutionary Science and Technology, Japan Science and Technology Agency) (Grant Number JPMJCR18T4) and JPJSBP-120219942 (X.H.).

## Author contributions

H.C. and X.H. conceived the idea, supervised the project, and wrote the manuscript. X.-X.W. and X.H. performed the theoretical analyses. Z.G. prepared the sample and conducted experimental measurements and the full-wave simulations. J.S. and H.J. helped with experiments. All authors fully contribute to the research.

## Competing interests

The authors declare no competing interests.
