## [Peer review file · Nature Communications]

REVIEWER COMMENTS

Reviewer #1 (Remarks to the Author):

This manuscript reports experiments on a microwave mesh with a Keluke honeycomb lattice which provides two phases, one where the character of the bands is preserved through the Brillouin zone (lattice elements in the unit cell are contracted as compared to the honeycomb) and one with a band inversion at the Gamma point (expanded). While this system is well studied, the reported findings are novel and seem intriguing.

However, I do not think the main claims are supported by the theory and experiments. My main concerns are two. First, regarding the theory, I do not find convincing the claim that the origin of the effect is in a non-trivial Berry curvature. Second, I have concerns regarding the experimental source being a chiral point source located at the centre of the unit cell, which is an essential assumption in the argument. Furthermore, I do not find convincing the analogy of the effect with negative refraction. Here I provide more explanations to support my comments.

1. Chiral coupling is governed by the spin angular momentum of the electromagnetic modes of the structure, which is related to the phase of the out of plane electric field and describes the local handedness of the field. This in turn determines the ability of sources to selectively excite modes with a specific handedness, such as directional modes. It has been reported in the literature that in this lattice the spin angular momentum changes sign between the centre and the edges of the unit cell, meaning that sources positioned in these different areas excite modes propagating in different directions, as for instance happens with the edge modes, see for instance Parappurath, N., Alpegiani, F., Kuipers, L., & Verhagen, E. (2020). Direct observation of topological edge states in silicon photonic crystals: Spin, dispersion, and chiral routing. *Science advances*, 6(10), eaaw4137, or Proctor, M., Craster, R. V., Maier, S. A., Giannini, V., & Huidobro, P. A. (2019). Exciting pseudospin-dependent edge states in plasmonic metasurfaces. *ACS Photonics*, 6(11), 2985-2995. It might be the case that the effect seen is the bulk counterpart of the edge mode directionality studied already. However the manuscript does not give results on the spin angular momentum obtained from first principles, or on any on any other quantities related to electromagnetic chirality, which would provide clear understanding of the effect.

2. Instead, they give an interpretation in terms of the Berry phase of the system which they relate to the orbital angular momentum. However the derivation might be problematic as it leads to some surprising features. Figure 2 e,f present plots of the calculated Berry curvature around the Gamma point, showing monopoles of opposite sign for the two phases of the system (trivial and inverted). Since time reversal symmetry is broken I wonder how this Berry curvature is compensated, do the plots correspond to just a subspace? There are two degenerate modes at the Gamma point such that a calculation of the non-Abelian Berry phase would be necessary but this is not performed. The obtained Berry curvature seems

to contradict the fact that it has been shown from first principles that this system does not have a Z2 topological invariant (see e.g. Blanco de Paz, M., Devescovi, C., Giedke, G., Saenz, J. J., Vergniory, M. G., Bradlyn, B., ... & García-Etxarri, A. (2020). Tutorial: computing topological invariants in 2D photonic crystals. *Advanced Quantum Technologies*, 3(2), 1900117.). There might be a flaw in the assumptions of first order perturbation theory. Is this correct in this system given the symmetry breaking away from the Gamma point?

3. From all this I wonder if there is indeed a change in chirality and if it could be the case that the fact that the source is not an actual point source is at the origin of the apparent change in chirality. As far as I can tell from the experimental details it looks like the mesh elements themselves are acting as sources, so they would not represent a point source located at the centre of the unit cell. This could mean that in the contracted and expanded phases they are exciting modes of different chirality because the spin angular momentum is of opposite sign in those regions of the unit cell. Since there are no details in how the simulations are performed it is difficult to assess if there could indeed be any finite size effects of the source, but in any case this should be ruled out.

4. Finally, there is no support to the claim that this could be an analogue to negative refraction. What is exactly the relation between the fact that the lower bands have a negative slope and the change of chirality seen here? The lower bands have negative slope in both phases and the effect is only seen in one. On the other hand, the upper bands have positive slopes and presumably the same physics would occur there.

Reviewer #2 (Remarks to the Author):

The authors study a topological photonic system in the RF regime in this paper. The bulk topology follows the model in the well-known and pioneering work [1], which presents a Z2 topology in a dielectric photonic crystal platform. The topological transition happens via effectively shrinking and expanding the lattice. In this work, the authors construct an RF photonic lattice with microstrips. An abnormal EM transportation is observed when a topologically non-trivial lattice is excited with a chiral source. Energy transports reversely around the lattice according to the source. The authors claim this is a novel Huygens-Fresnel electromagnetic transportation.

Personally, I understand that this work is trying to obtain some observable physical consequences of the topology of the photonic bandstructure. However, I feel that the story is not well-built and the work is not well presented. The main claim about the novel Huygens-Fresnel is ambiguous and somehow

misleading. Thus, I'm sorry that this work doesn't add to our understanding of this area and it doesn't meet the standard of Nature Communications.

First, I wonder whether the Huygens-Fresnel principle is a good choice to analyze this system. Usually, people talk about it in the far-field limit situation. However, the platform in this work is the extreme opposite. The whole system can be treated as a lumped circuit and the measurement is done with near-field probing. Besides, as far as I understand, people talk about the movement of the phase fronts in this situation, which doesn't have to be the same with the Poynting vector.

Second, I'm not sure whether this transportation is really novel. Similar transportations, in which energy flows in the opposite direction of the phase propagation direction, can happen in many photonic periodic structures near the band edges.

Third, as far as I understand, the authors are trying to find observable physical consequences of the bulk band topology. Mapping between band structures, mode profiles at high-symmetry points, and far-field radiation patterns always exist, as shown in some early works in vector beam generation. I wonder whether this platform has many advantages over those such as dielectric photonic crystal slabs, metal rod arrays for RF photonics, etc. Personally, I do feel studying the energy flow in a unit cell of a periodic structure is not straightforward.

Finally, I have a few technical questions.

As pointed out by the authors, this transportation is fully defined by the bulk topology. Then, what's the meaning of topologically different cladding? The discussion on noise is not very convincing. Personally, I feel that a large enough pure periodic structure is more straightforward.

I'm still not sure about the relationship between the transportation and the bulk band topology. First, the topological transition can be understood by redefining the unit cell. Thus, boundaries are necessary for the real structure to make the topology well defined. Then, what if we make a large enough structure and then shift the relative position between the chiral source and the structure? Second, the same trick can be used to change the spatial symmetry indices of modes at high symmetric points in the momentum space. Will it have the same effect without this specific Z2 topology?

[1] Wu, Long-Hua, and Xiao Hu. "Scheme for achieving a topological photonic crystal by using dielectric material." *Physical review letters* 114.22 (2015): 223901.

Reviewer #1 (Remarks to the Author):

This manuscript reports experiments on a microwave mesh with a Keluke honeycomb lattice which provides two phases, one where the character of the bands is preserved through the Brillouin zone (lattice elements in the unit cell are contracted as compared to the honeycomb) and one with a band inversion at the Gamma point (expanded). While this system is well studied, the reported findings are novel and seem intriguing.

We are grateful to this Reviewer for spending time to review our manuscript and the assessment on our work.

However, I do not think the main claims are supported by the theory and experiments. My main concerns are two. First, regarding the theory, I do not find convincing the claim that the origin of the effect is in a non-trivial Berry curvature. Second, I have concerns regarding the experimental source being a chiral point source located at the centre of the unit cell, which is an essential assumption in the argument. Furthermore, I do not find convincing the analogy of the effect with negative refraction. Here I provide more explanations to support my comments.

We thank this Reviewer for raising valuable questions, which we are going to address in what follows. In order to resolve the first, theoretical concern, we shall provide detailed explanations on the role of Berry phase when frequency is set in passband. As for the second, experimental concern, we notice that the chiral source is not required to be strictly point like, which is unrealistic in any sense, and we have not assumed this in our theory. Notice that the spinor wave function associated with the Dirac Hamiltonian is defined on the C_{6v} -symmetric real-space unit cell, rather than at a single point. Regarding the well-known negative refraction, we did not take advantage of the possible analogy to clarify the novel chiral response in the present work. Having in mind that one is for plane wave and the other for chiral wavelet source and that both properties appear around the band edges, we consider they are counterpart to each other and wish to highlight the usefulness of their combination for future photonic applications. In what follows we address the concerns of this Reviewer in a point-by-point fashion. The comments of this Reviewer really help us sharpen the presentation of our work.

1. Chiral coupling is governed by the spin angular momentum of the electromagnetic modes of the structure, which is related to the phase of the out of plane electric field and describes the local handedness of the field. This in turn determines the ability of sources to selectively excite modes with a specific handedness, such as directional modes.

We totally agree with this Reviewer on this point.

It has been reported in the literature that in this lattice the spin angular momentum changes sign between the centre and the edges of the unit cell, meaning that sources positioned in these different areas excite modes propagating in different directions, as for instance happens with the edge modes, see for instance Parappurath, N., Alpeggiani, F., Kuipers, L., & Verhagen, E. (2020). Direct observation of topological edge states in silicon photonic crystals: Spin, dispersion, and chiral routing. Science advances, 6(10), eaaw4137, or Proctor, M., Craster, R. V., Maier, S. A., Giannini, V., & Huidobro, P. A. (2019). Exciting pseudospin-dependent edge states in plasmonic metasurfaces. ACS Photonics, 6(11), 2985-2995.

We humbly disagree with this Reviewer on this point, noticing that in the present design the chiral response depends crucially on whether the frequency is set in the passband or falls in the bandgap, and show evidence which supports our statement in the following. As displayed in Figure I and Figure II, for a passband frequency under concern the whirling direction of local Poynting vectors in C_{6v} -symmetric unit cells, which specifies the pseudospin state, is the same as the chiral source, irrespective to the location of the chiral source, either in the C_{6v} -symmetric unit cells (shown in manuscript) or in the C_{3v} -symmetric area among three C_{6v} -symmetric unit cells.

Fig. I: Chiral response in the trivial structure to a ccw chiral source located in C_{3v} -symmetric area with frequency below the lower band edge. The small hexagonal structure at the center is the source. The system with 108 C_{6v} -symmetric unit cells is exposing to air without cladding.

Fig. II: Chiral response in the topological structure to a ccw chiral source located in C_{3v} -symmetric area with frequency below the lower band edge. All conditions are put same as in Fig. I.

In contrast, for in-gap frequency as discussed in the two references cited by this Reviewer,

the whirling direction of local Poynting vectors in C_{6v} -symmetric unit cells is the same as the chiral source only when the source is located in a C_{6v} -symmetric unit cell, as shown in Figure III (a) and (b), while it is opposite to the source when the source is located in the C_{3v} -symmetric area among three unit cells, as shown in Figure III (c) and (d). Simulation results in Figures I, II and III have been obtained under the same conditions except the frequency, and those in Figure III are in good agreement with the results reported in the paper by M. Proctor et al. in ACS Photonics (2019). As for the paper by N. Parappurath et al. in Science Advances (2020) (Ref. #26 in our manuscript), we notice that the size of incident light spot from above to PhC is larger than the unit cell, and thus it is inconvenient for discussion on the effect of source position either inside a C_{6v} -symmetric unit cell or in the C_{3v} -symmetric area among three unit cells. Instead, N. Parappurath et al. showed clearly that the topological interface mode is along the same direction no matter the chiral source is located in topological region or trivial region, as far as the source frequency falls in the band gap.

Fig. III: Pseudospin states induced by a ccw chiral source with frequency in the band gap: (a) & (b) chiral source located in the C_{6v} -symmetric unit cell for the trivial structure and the topological structure, respectively, which induces pseudospin parallel to the source; (c) & (d) chiral source located in the C_{3v} -symmetric area for the trivial structure and the topological structure, respectively, which induces pseudospin opposite to the source.

It might be the case that the effect seen is the bulk counterpart of the edge mode directionality studied already.

As demonstrated above, the location of chiral source, i.e. in a C_{6v} -symmetric unit cell and/or in the C_{3v} -symmetric area among unit cells, leaves different impacts on the mode

excitation when the frequency is set in passband and/or falls in bandgap. It is clear that results for in-gap frequency cannot be used to infer properties for frequency in the passband.

In the present work, we mainly concentrate on the case that the chiral source is located in a C_{6v} -symmetric unit cells since it is easy to achieve in experiments, and the frequency is exclusively set in the passband. The phenomenology remains unchanged even the source is shifted to the C_{3v} -symmetric area among three unit cells: the global bulk EM flow is opposite to the chiral source in topological PhC whereas it is the same in trivial PhC (see Figure I and Figure II). In contrast, the topological interface transportation induced by a chiral source with in-gap frequency is along the same direction no matter the source is located in the topological region or the trivial region, as observed in experiments reported in the paper by N. Parappurath et al. in Science Advances (2020) (Ref. #26 in our manuscript) with the Jackiw-Rebbi origin associated with two opposite Dirac masses (see for example Ref. #32 of our manuscript).

In short, the effect reported in the present manuscript is a novel phenomenon, which has a new physical origin and cannot be understood in terms of the directionality of the topological interface mode. The key difference lies in the frequency.

To be complete, in the new version of our manuscript we add in Supplementary Materials simulation results for the cases when the chiral source is located in the C_{3v} -symmetric area among three unit cells, and those for frequency in band gap. In order to accommodate the comment of this Reviewer, we add discussions on the difference between the cases for frequency in passband and in bandgap, where we cite additionally the paper by M. Proctor et al. in ACS Photonics (2019) (the paper by N. Parappurath et al. in Science Advances (2020) has been cited already). For details please see the revised manuscript.

However the manuscript does not give results on the spin angular momentum obtained from first principles, or on any other quantities related to electromagnetic chirality, which would provide clear understanding of the effect.

It is not the case. In the previous version of our manuscript, the phase winding, thus pseudospin state or equivalently chirality, of EM modes excited in C_{6v} -symmetric unit cells by a ccw source has been displayed in Fig. 4 (f) obtained by “first-principle”, full-wave simulations using CST software package on the trivial structure. The corresponding results are summarized in Fig. 5(f) for the topological structure. They are in complete agreement with experimental results as shown in Figs. 4(b) and 5(b), and in full agreement with our theory as documented in the manuscript.

2. Instead, they give an interpretation in terms of the Berry phase of the system which they relate to the orbital angular momentum. However the derivation might be problematic as it leads to some surprising features. Figure 2 e,f present plots of the calculated Berry curvature around the Gamma point, showing monopoles of opposite sign for the two phases of the system (trivial and inverted). Since time reversal symmetry is broken I wonder how this Berry curvature is compensated, do the plots correspond to just a subspace? There are two degenerate modes at the Gamma point such that a calculation of the non-Abelian Berry phase would be necessary but this is not performed.

The Berry phases shown in Fig. 2 are for the case of a ccw chiral source, as indicated by the curved arrow at the center unit cell, which breaks the time-reversal symmetry; all the plots, including Fig. 2 (e,f), correspond to the subspace of up-pseudospin picked up by the ccw source. The criticism “the derivation might be problematic as it leads to some surprising features” will be addressed in the next rebuttal.

*The obtained Berry curvature seems to contradict the fact that it has been shown from first principles that this system does not have a Z2 topological invariant (see e.g. Blanco de Paz, M., Devescovi, C., Giedke, G., Saenz, J. J., Vergniory, M. G., Bradlyn, B., ... & García-Etxarri, A. (2020). Tutorial: computing topological invariants in 2D photonic crystals. *Advanced Quantum Technologies*, 3(2), 1900117.). There might be a flaw in the assumptions of first order perturbation theory. Is this correct in this system given the symmetry breaking away from the Gamma point?*

First, please let us notice that the absence (or existence) of a Z2 topological invariant index in the distorted honeycomb structures is irrelevant to the physics reported in the present work, since (1), as discussed above, the time-reversal symmetry is broken by the chiral source and only one pseudospin subspace is picked up; (2) the bulk transportation property under concern in the present work is for frequency in passband close to the band edge, which leaves the Berry curvatures in the remaining part of BZ away from Γ point irrelevant. In contrast, the Z2 topological invariant index derived by integrating Berry curvatures over the whole BZ, like other topological invariant index such as Chern number, is important when one considers possible in-gap topological interfacial states according to the bulk-edge correspondence.

As for the absence (or existence) of the Z2 topological invariant index in the Kekule-distorted honeycomb PhC, we confirmed the paper by M. Blanco de Paz et al. which suggests its absence. However, we have to say that that work is hardly taken as “from first principles”. As a matter of fact, there is a latter paper by S. J. Palmer and V. Giannini in

Phys. Rev. Res. (2021) (Ref.#35 in our manuscript), where its existence is suggested, opposite to the statement of M. Blanco de Paz et al., and furthermore the discrepancy between the two approaches has been discussed in details. Therefore, the absence of Z2 topological invariant index in the present structure is not a “fact”.

Therefore, we cannot find concrete support for the following criticisms raised by this Reviewer “the derivation might be problematic as it leads to some surprising features” and “there might be a flaw in the assumptions of first order perturbation theory”. From the fine agreement among results from theory, full-wave computer simulations and experiments on the novel chiral response in the topological structure, it is reasonable to say that the theoretical treatment based on the first order perturbation is valid for the present physics.

In order to accommodate the comment of this Reviewer, we add a discussion on the absence/existence of the Z2 topological invariant index in the present structure and its irrelevance to the present property, where we add the paper by M. Blanco de Paz et al. in our reference list. Meanwhile, we soften our statement about the Berry-phase origin of the phenomenon. For details, please see the new manuscript.

3. From all this I wonder if there is indeed a change in chirality and if it could be the case that the fact that the source is not an actual point source is at the origin of the apparent change in chirality. As far as I can tell from the experimental details it looks like the mesh elements themselves are acting as sources, so they would not represent a point source located at the centre of the unit cell.

As addressed above and summarized in Figure I and Figure II and in the manuscript, the global EM flow is opposite to the chiral source in the topological structure no matter the chiral source is located in a C_{6v} -symmetric unit cell or in the C_{3v} -symmetric area among three unit cells. There is no requirement that the chiral source should be a point-like one, which is unrealistic in any way, noticing that the spinor wave function associated with the Dirac Hamiltonian is defined on the real-space C_{6v} -symmetric unit cell, rather than a single point. Instead, the whirling direction of local Poynting vector is the central feature, which picks up one of the two pseudospin states in the system specified by the whirling direction of EM flow in individual C_{6v} -symmetric unit cells. Our theory, full-wave simulations and experiments are all in good agreement, and indicate coherently the novel property in topological PhC in the passband.

We regretfully find that on line 6 of page 3 of our previous manuscript there was a description on “chiral point-like source” in the context of topological interface EM

transportation, which might have caused unnecessary confusions. It is incorrect since sources reported in most literatures are of sizes comparable to or even larger than the unit cells. A similar expression was used in the caption of Fig. S23 (Fig. S14 in the previous version). In order to be exact, we change them into “chiral source” in the new version of manuscript.

This could mean that in the contracted and expanded phases they are exciting modes of different chirality because the spin angular momentum is of opposite sign in those regions of the unit cell. Since there are no details in how the simulations are performed it is difficult to assess if there could indeed be any finite size effects of the source, but in any case this should be ruled out.

It is not the case. As addressed above and summarized in Figure I and Figure II (see also Figs. 4 and 5 in the main text of our manuscript), for frequency in passband the EM modes excited by a chiral source always carry the chirality (which is specified by the whirling direction of EM flow in individual C_6 -symmetric unit cells) same to the source, no matter the structure is topological (expanded) or trivial (contracted), no matter the source is located in the center of a C_{6v} -symmetric unit cell or in the C_{3v} -symmetric area among three unit cells. The size effect of source is irrelevant as far as it is contained in one unit cell.

The experimental details have already been shown in Fig. S5. In order to resolve the concern of this Reviewer, we add detailed explanations on the full-wave computer simulations on the source. As an input port we apply a voltage of 1V between a node on the top honeycomb structure and the corresponding point on the bottom conductor of the microstrip line platform, and construct a chiral source by decreasing the phase of voltage clockwise (counterclockwise) between neighbor nodes in a C_{6v} and/or C_{3v} -symmetric hexagon.

4. Finally, there is no support to the claim that this could be an analogue to negative refraction. What is exactly the relation between the fact that the lower bands have a negative slope and the change of chirality seen here? The lower bands have negative slope in both phases and the effect is only seen in one. On the other hand, the upper bands have positive slopes and presumably the same physics would occur there.

It has been established for a while that negative refraction appears for frequency below the lower band edge at Γ point realized in a PhC (see Refs. #8 and #9 for the pioneering theoretical and experimental works in this field), since, as mentioned by this Reviewer, it

is induced by the negative band mass associated with the upwardly convex dispersion relation. Therefore, negative refraction is realized in both topological and trivial structures as far as the frequency in the lower passband, as displayed in Figure IV.

In contrast, the novel chiral response clarified in the present work is induced by the negative Dirac mass M as realized in the topological (expanded) structure, both below and above the bandgap around the band edges.

Fig. IV: Negative refraction in the trivial (left) and topological (right) photonic structures for frequency below the lower band edge.

In negative refraction realized in photonic structures, the parallel component of the wave momentum of refracted plane wave is opposite to the incident one, while in the novel chiral response addressed in the present work the global EM flow is opposite to the chiral source. In this sense, we consider them as counterpart to each other. Because they both happen in passbands close to the band gap of PhC, there are four possible cases for the refraction and the chiral response as summarized in Fig. S23, which may be exploited for realizing advanced photonic manipulation in the future. It is clear that through the work we do not take advantage of the possible analogy between these two phenomena in analysis.

In order to accommodate the comment from this Reviewer, we add discussions on this point in the main text of manuscript. For details please see the new manuscript.

Finally, we wish to express our heartfelt thanks to this Reviewer for spending valuable time to review our work, give nice overall assessment and raise important comments. The course of addressing the criticisms indeed helps us sharpen our presentation. With all the comments addressed, we sincerely hope that we can obtain from this Reviewer recommendation for publication of our work in Nature Communications.

Reviewer #2 (Remarks to the Author):

The authors study a topological photonic system in the RF regime in this paper. The bulk topology follows the model in the well-known and pioneering work [1], which presents a Z2 topology in a dielectric photonic crystal platform. The topological transition happens via effectively shrinking and expanding the lattice. In this work, the authors construct an RF photonic lattice with microstrips. An abnormal EM transportation is observed when a topologically non-trivial lattice is excited with a chiral source. Energy transports reversely around the lattice according to the source. The authors claim this is a novel Huygens-Fresnel electromagnetic transportation.

We are grateful to this Reviewer for spending time to review our manuscript, and the evaluation on our work.

Personally, I understand that this work is trying to obtain some observable physical consequences of the topology of the photonic band structure. However, I feel that the story is not well-built and the work is not well presented. The main claim about the novel Huygens-Fresnel is ambiguous and somehow misleading. Thus, I'm sorry that this work doesn't add to our understanding of this area and it doesn't meet the standard of Nature Communications.

We thank this Reviewer for raising critical comments on our work. In what follows we address them in a point-by-point fashion. The criticisms really help us improve the presentation of our results.

First, I wonder whether the Huygens-Fresnel principle is a good choice to analyze this system. Usually, people talk about it in the far-field limit situation. However, the platform in this work is the extreme opposite. The whole system can be treated as a lumped circuit and the measurement is done with near-field probing. Besides, as far as I understand, people talk about the movement of the phase fronts in this situation, which doesn't have to be the same with the Poynting vector.

As observed in our experiments and full-wave computer simulations, a chiral source located in the center C_{6v} -symmetric unit cell induces whirling EM flows in neighboring unit cells in the same direction, which is determined by the phase winding and specifies the pseudospin state; in turn these unit cells induce the same chiral states in outer unit cells, and so on so forth. Namely all unit cells can be considered as secondary sources with the same chirality, a situation rendered uniquely by the Dirac Hamiltonian Eq. (1).

In addition, the Dirac mass M (including its sign) and the Bloch wavenumber in the chiral

form k_{\pm} govern the phase interferences both inside and between unit cells in such the special way as given in Eq. (3), which results in a global EM flow in the topological PhC ($M < 0$) opposite to the chiral source.

In our opinion, these new findings in the photonic Dirac system add new facet to the Huygens-Fresnel principle, having in mind that in Huygens-Fresnel picture people understand the destination of electromagnetic wave in terms of phase interferences among stimulated, secondary sources. Needless to say, through the whole work our analyses are “microscopic” and do not rely on any simplified picture.

We also wish to put it clear that in the present work we explore the peculiar bulk EM transportation in a system characterized by the Dirac-type frequency dispersion and the associated spinor wavefunction, which is inappropriate to be categorized in terms of far-field and/or near-field limit, although the measurement of local electric field in the microstrip structure belongs to a near-field probe.

Second, I'm not sure whether this transportation is really novel. Similar transportations, in which energy flows in the opposite direction of the phase propagation direction, can happen in many photonic periodic structures near the band edges.

We humbly disagree with this Reviewer on this point. As mentioned by this Reviewer, it has been established for a while that negative refraction appears for frequency below the lower band edge at Γ point realized in a photonic crystal (see Refs. #8 and #9 for the pioneering theoretical and experimental works). This negative refraction is induced by the negative band mass associated with the upwardly convex dispersion relation, and appears in both topological and trivial structures, as displayed in Figure IV.

In contrast, the new chiral response clarified in the present work takes place when the band gap is characterized by a negative Dirac mass M as achieved in the topological (i.e. expanded) structure, and appears both below and above the photonic bandgap exclusively in the topological structure.

Therefore, although both related to the frequency band structures, these two phenomena take place with completely different mechanisms, and are associated with plane wave and chiral wavelet respectively, which drives us to consider this never-before-seen chiral response as a novel EM transportation.

In order to accommodate this comment we add paragraphs to emphasize the similarity and difference between the two phenomena. For details please see the new version of manuscript.

Third, as far as I understand, the authors are trying to find observable physical consequences of the bulk band topology. Mapping between band structures, mode profiles at high-symmetry points, and far-field radiation patterns always exist, as shown in some early works in vector beam generation. I wonder whether this platform has many advantages over those such as dielectric photonic crystal slabs, metal rod arrays for RF photonics, etc. Personally, I do feel studying the energy flow in a unit cell of a periodic structure is not straightforward.

In the present work we concentrate on bulk EM transportations in honeycomb-type photonic structures, which are already very rich on their own right due to the Dirac physics unable to access directly in experiments in any other systems so far. While the far-field radiation should be intriguing as well, it is not the main concern here.

The direction of energy flow in a real-space C_{6v} -symmetric unit cell specifies the chirality or pseudospin state in the present photonic Dirac system. Thus, in order to clarify the Dirac physics, it is crucially important to prepare a chiral source which excites accurately the mode with specified chirality in the system and to check the energy flow in individual unit cells directly in terms of near-field experimental techniques. As the first proof-of-concept experiment, the microstrip is the best choice since it can be precisely excited by a chiral source with specific phase winding and, thanks to the completely flat structure and the working RF frequency, one can measure the real-space distributions of both phases and amplitudes of the out-plane electric field with high precision, which are used to evaluate the energy flow, both inside unit cells and global. In contrast, the near-field information cannot be obtained easily in photonic crystals due to much smaller real-space structures associated with higher frequency, and in RF metal rod arrays due to much sharper variations in height landscape.

As for vector beam generation based on photonic crystal slabs mentioned by this Reviewer, relatively well understood near-field properties are exploited for achieving advanced far-field properties such as vortex beams (see for example B. Wang et al., Nat. Photon. vol. 14, 623 (2018)). While both approaches are making use of momentum-space properties, there are obvious differences between them: (1) in photonic crystal slabs the thickness should be well controlled, which builds a singlet frequency dispersion, involving both TM and TE guide modes for example, whereas in our system the Dirac dispersion is controlled by the in-plane photonic structure with the thickness being irrelevant, where only one of the TM and TE modes works and rich physics emerges from the interweaving between p-mode and d-mode with opposite parity respective to spatial reversal; (2) inside photonic crystal slabs all photonic states are linear polarized, and the

vortex beam radiation is induced by a focused incident light with circular polarization, whereas in our system the doubly degenerate photonic eigenstates are chiral associated with the C_{6v} symmetry of the honeycomb photonic structure, and the chiral source picks up one of them. At this moment, we certainly cannot claim that our approach is advantageous as compared the vector beam generation for generating vortex beam, and wish to leave detailed comparison between them as an intriguing future issue.

Finally, I have a few technical questions. As pointed out by the authors, this transportation is fully defined by the bulk topology. Then, what's the meaning of topologically different cladding? The discussion on noise is not very convincing. Personally, I feel that a large enough pure periodic structure is more straightforward.

The cladding layer is indeed not a must as pointed out by this Reviewer. In Figure V and Figure VI, we show full-wave computer simulations for a large pure system with 127 C_{6v} -symmetric unit cells exposing to air, either of topological or trivial structure, under a ccw chiral source. It is clear that our results remain unchanged.

We thank this Reviewer for this comment, and include these simulation results and those for cw chiral source in Supplementary Materials.

Fig. V: Chiral response in the trivial structure without cladding.

Fig. VI: Chiral response in the topological structure without cladding.

I'm still not sure about the relationship between the transportation and the bulk band topology. First, the topological transition can be understood by redefining the unit cell. Thus, boundaries are necessary for the real structure to make the topology well defined. Then, what if we make a large enough structure and then shift the relative position between the chiral source and the structure?

We humbly disagree with the Reviewer on that the topological transition can be understood by redefining the unit cell. The topology in the present system relies on the C_{6v} symmetry. This Z2 topology is defined without referring to a boundary in real space, unlike the 1D SSH model whose topology is characterized by a winding number. The topology transition takes place when the structure is modified from expanded to shrunken via the perfect honeycomb structure with zero band gap, where during the whole course the C_{6v} symmetry should be always respected. While taking the unit cell in a way respecting C_{3v} symmetry would not change the band structure, no topology can be defined anymore due to the reduction of symmetry.

On the other hand, it is a valid question to ask what if one makes a large enough structure and then shifts the relative position between the chiral source and the structure (even though this manipulation is not related directly to transition of topology as addressed above). Following the suggestion from this Reviewer, we have performed additionally full-wave computer simulations with the results displayed in Figure I and Figure II. It is clear that, with the chiral source shifted to the C_{3v} -symmetric area among three unit cells, the chiral response remains unchanged provided the source frequency is set in the passband.

We thank this Reviewer for raising the comment, and include the new simulation results in the Supplementary Materials.

Second, the same trick can be used to change the spatial symmetry indices of modes at high symmetric points in the momentum space. Will it have the same effect without this specific Z2 topology? [1] Wu, Long-Hua, and Xiao Hu. "Scheme for achieving a topological photonic crystal by using dielectric material." Physical review letters 114.22 (2015): 223901.

As displayed in Figure 1, shifting the chiral source from the C_{6v} -symmetric unit cell to the C_{3v} -symmetric area among three unit cells does not change the chiral response, as far as the frequency is set in passband which is the main concern of the present work. At this moment, it is unclear whether similar chiral response takes place in other systems without the Z2 topology associated with honeycomb structure, which is beyond the scope of the present work but certainly constitutes an intriguing future problem.

Finally, we wish to express our heartfelt thanks to this Reviewer for spending valuable time to review our work and raise important comments. The course of addressing the

criticisms indeed helps us sharpen our presentation. With all the comments addressed, we sincerely hope that we can obtain from this Reviewer recommendation for publication of our work in Nature Communications.

Rebuttal to the second report of Reviewer #1

(C0) *I thank the authors for their extensive reply. Unfortunately I still find some of the claims are not well supported. Furthermore, the main points are the link to topological properties and negative refraction, and these are still unclear, such that the significance and impact of the work in the field are unclear.*

(R0) We are thankful to Reviewer #1 for spending time to review our rebuttal and revised manuscript. For those Reviewer #1 still finds not well supported, we reply in what follows in a point-to-point fashion.

Before going to the details, we would like to respond to the comment of Reviewer #1 on the comparison between topological chiral response and negative refraction. As demonstrated in our manuscript and put in our last rebuttal report, (1) both topological chiral response and negative refraction occur at frequencies close to band edges and referring to bulk optic transportation; (2) the former is dominated by the negative Dirac mass while the latter is induced by the negative band mass; (3) the former involves circular polarization and orbital angular momentum (OAM) in the normal direction while the latter governs the in-plane propagation of linearly polarized light. Therefore, the novel chiral response and negative refraction are of different physical origins and provide complementary features.

Also the answer is given hereby to the comment of Reviewer #1 raised in the second review report about the significance and impact of our present work: The present work highlights novel **bulk optical features** associated with the pseudospin and global OAM fostered in the honeycomb structure. Opposed to waveguide-like topological **interface modes** investigated intensively so far, they are compatible to a broader scope of optical applications, including sensor, filter, microscopy, as well as tweezer and spanner for particle manipulation [1-8]. Superior to defect-type structures [1, 9], the periodic lattice structure in the present approach can be developed to generate multi-singularity optical vortices useful for high-resolution far-field microscopy [10]. The photonic lattice structure also reconciles with lateral propagation of plane wave, which exhibits negative refraction at interfaces to uniform media with frequency below the lower band edge [11, 12].

In order to accommodate these comments from Reviewer #1, we revise the manuscript and put these points in a clearer way and adding relevant previous works in the reference list. For details please see the revised manuscript.

References:

1. Shen, Y., Wang, X., Xie, Z., Min, C., Fu, X., Liu, Q., Gong, M., Yuan, X. Optical vortices 30 years on: OAM manipulation from topological charge to multiple singularities. *Light Sci Appl* **8**, 90 (2019).
2. Shao, Z. K., Chen, H. Z., Wang, S., Mao, X. R., Yang, Z. Q., Wang, S. L., Wang, X. X., Hu, X. & Ma, R. M. A high-performance topological bulk laser based on band-inversion-induced reflection. *Nat. Nanotechnol.* **15**, 67-72 (2020).
3. Gao, P., Christensen, J. Topological vortices for sound and light. *Nat. Nanotechnol.* **16**, 487-489 (2021).
4. Li, T., Xu, X., Fu, B., Wang, S., Li, B., Wang, Z., Zhu, S. Integrating the optical tweezers and spanner onto an individual single-layer metasurface. *Photonics Res.* **9**, 1062 (2021).
5. Hakobyan, D., Brasselet, E. Left-handed optical radiation torque. *Nat. Photon.* **8**, 610-614 (2014).
6. Magallanes, H., Brasselet, E. Macroscopic direct observation of optical spin-dependent lateral forces and left-handed torques. *Nat. Photon.* **12**, 461–464 (2018).
7. Shi, Y., Zhu, T., Liu, J., Tsai, D. P., Zhang, H., Wang, S., Chan, C. T., Wu, P. C., Zayats, A. V., Nori, F., Liu, A. Q. Stable optical lateral forces from inhomogeneities of the spin angular momentum. *Sci. Adv.* **8**, eabn2291 (2022)
8. Shi, Y., Zhu, T., Zhang, T. et al. Chirality-assisted lateral momentum transfer for bidirectional enantioselective separation. *Light Sci Appl* **9**, 62 (2020).
9. Devlin, R. C., Ambrosio, A., Rubin, N. A., Mueller, J. P. B., Capasso, F. Arbitrary spin-to-orbital angular momentum conversion of light. *Science* **358**, 896 (2017).
10. Xie, X., Chen, Y., Yang, K., Zhou, J. Harnessing the point-spread function for high-resolution far-field optical microscopy. *Phys. Rev. Lett.* **113**, 263901 (2014).
11. Notomi, M. Theory of light propagation in strongly modulated photonic crystals: Refractionlike behavior in the vicinity of the photonic band gap. *Phys. Rev. B* **62**, 10696 (2000).
12. Cubukcu, E., Aydin, K., Ozbay, E., Foteinopoulou, S. & Soukoulis, C. M. Negative refraction by photonic crystals. *Nature* **423**, 604-605 (2003).

(C1) *First, I thank the authors for clarifying that the position of the chiral source does not have any effect in propagation direction when the frequency is within the band. I agree it is now clear that there is a change in rotation direction when comparing the contracted and expanded lattices. This can also be understood by looking at the areas of the unit cell where the field is more intense. As a minor note the Poynting vector plots are not too clear due to poor resolution.*

(R1) We are happy that now Reviewer #1 agrees with us on the main observation of the present work and accepted our description on the novel chiral response due to the Dirac physics in the expanded lattice. We apologize for poor resolution of figures which might occur due to duplication, while the original figures are sufficiently clear.

(C2) *On the other hand, I still find some of the claims in the paper are not well supported. The authors reply “The Berry phases shown in Fig. 2 are for the case of a ccw chiral source, as indicated by the curved arrow at the center unit cell, which breaks the time-reversal symmetry; all the plots, including Fig. 2 (e,f), correspond to the subspace of up-pseudospin picked up by the ccw source.” However the Berry curvature and the Berry phase are a property of the eigenstates of the system Hamiltonian, and for that reason it cannot correspond to “the case of a ccw chiral source” as claimed by the authors. Hence it is not clear to me what is calculated in particular.*

(R2) We thank Reviewer #1 for raising this question. In the present work, we focus on the bulk chiral response of the system modes in the vicinity of band edges. As given in the $k \cdot p$ theory to the lowest order of momentum k in Eq. (1), the Hamiltonian is decoupled into two subspaces governed by opposite pseudospins. Encouraged by the appreciation of perturbation approach by Reviewer #1 as dictated in the second review report (see C6), we have performed the second-order perturbation analysis with additional higher orders of momentum k in the $k \cdot p$ Hamiltonian, where the two pseudospin sectors are mixed (see Eq. (A24) in the Supplementary Note III). It is shown clearly that the wavefunction excited by ccw (cw) chiral sources take the pseudospin-up (-down) eigenstate given in the lowest-order theory. Therefore, the second-order perturbation theory justifies in a transparent way the physical picture based on the first-order perturbation analysis. In order to enhance the presentation of core picture, we mention the above physics in the main text of manuscript and leave the whole second-order perturbation analysis in the Supplementary Note III. All these said, the plots shown in Fig. 2 (e,f) in the manuscript correspond to the Berry

curvature of the eigenmodes in the pseudospin-up subspace of the lowest-order Hamiltonian, which is excited by the ccw chiral source.

(C3) *As is clear and agreed by the authors, the system is time reversal symmetric and, while there can be accumulations of Berry curvature in some places of the Brillouin zone such as the ones plotted by the authors, the integral throughout the Brillouin zone has to be zero. This is contradictory with finding a non-zero Berry phase at the Gamma point.*

(R3) As addressed above, in the lowest-order $k \cdot p$ theory there are two pseudospin sectors, which are transferred to each other by time-reversal operation, yielding the time-reversal symmetry of the whole system. It is noticed that the individual sectors are not time-reversal symmetric, which permits non-zero Berry phase even at Γ point, a time-reversal invariant point in the Brillouin zone. Physically a pseudospinful state is excited by a chiral source as shown clearly by the second-order perturbation analysis (see Supplementary Note III, as well as C4 and R4).

(C4) *If this is being found is just because the calculation is selecting a given subspace of the Hamiltonian (“one pseudo-spin” as is often called), that is selected by the source. In my view this is what the authors refer to when saying “the case of a ccw source”, but this is not mathematically well founded.*

(R4) We are glad that Reviewer #1 captured our point. As addressed in the above rebuttals, we have formulated the second-order perturbation theory, which demonstrates analytically that chiral sources excite states with wavefunction coinciding with the pseudospinful eigenstates given by the lowest-order theory. For details, please refer to the Supplementary Note III.

(C5) *Figure 3 in the manuscript shows how the bands below and above the gap touch not only at the Gamma point but at other points, such that it has to be demonstrated that the manifold is indeed globally separable into two subspaces. Otherwise one needs to calculate a non-Abelian Berry curvature, but this is not the approach taken.*

(R5) This is an iterated comment from the last report. First of all, we wish to notice that, as documented in the comment C6 of Reviewer #1 and our rebuttal, the main physics in the present work is governed by the property around the Gamma point since the frequency is set slightly below

(above) the lower (upper) band edge. This is different from the case where the frequency is set in the band gap, for which the pseudospinful Berry phases accumulated in the whole Brillouin zone are important.

For Berry phases in the whole Brillouin zone, as addressed in the last rebuttal report, Palmer and Giannini [Phys. Rev. Res. vol. 3, L022013 (2021), Ref. 35 in our manuscript] have performed calculations on non-Abelian Berry curvatures for a photonic crystals with band structures same as the present ones, and demonstrated that the manifold is indeed globally separable into two subspaces, where the non-Abelian Berry curvature of the two-fold subspace can be diagonalized in terms of pseudospin. We cited the paper by Palmer and Giannini in the first version of our manuscript, and emphasized its relevance to our work and to the comment from Reviewer #1 in the last rebuttal report.

Just for convenience we list up the following results in Ref. 35: (1) the frequency band gap remains open when one changes the expanded structure based on circle pillars in Ref. 21 gradually to the elliptic pillars in Ref. 50 with C_{6v} symmetry respected all the way, see Fig. 4a in Ref. 35; (2) these two systems exhibit very similar interfacial modes when topological and trivial structures are put side-by-side in the ribbon geometry respectively, see Fig. 4b&c in Ref. 35. These calculations show clearly that the topology investigated in Ref. 50 is the same as that investigated in Ref. 21.

(C6) It looks like in this case the problem is indeed separable at least within perturbation theory and it looks like this enough to describe the bulk source problem, unlike the global topological properties of the system. All these mathematical details are important and not well discussed in the paper. For that reason I still find the connection to the Berry curvature is not well supported, and Fig 3 to be unclear.

(R6) We are happy that now Reviewer #1 appreciates the validness of the perturbation theory on the physics presented in our manuscript. In accordance to this, we present the second-order perturbation analysis and add it to the supplementary material, which shows that our picture based on the lowest-order theory in Eqs. (1) and (3) with two decoupled pseudospin sectors is valid for the discussion on chiral responses.

As detailed in Supplementary Note III, the second-order perturbation analysis starts from a 4x4 Dirac Hamiltonian in momentum space where terms mixing the two pseudospin sectors are

included. As is shown unambiguously there, the wavefunction excited by a ccw (cw) chiral source coincides with the pseudospin-up (-down) eigenstate given in the lowest-order theory. It becomes clear that, modifications in the spinor wavefunctions due to broken time-reversal symmetry off Γ point only appear in higher-order infinitesimals $O(k^3)$, which can be neglected safely as far as the physics around band edges is concerned as in the present work. Therefore, the physical picture based on the pseudospin in the scheme of the first-order perturbation documented in the manuscript is substantiated clearly. We add a paragraph to describe the second-order perturbation at the end of subsection “Theory” in the main text of manuscript. For details please see the manuscript.

It is worth noticing that around the Γ point the discussions of Ref. 35 match well with our second-order perturbation analysis, especially the ways how the pseudospin-up and -down states evolve into two non-degenerated dispersions coincide with each other in these two approaches.

(C7) *I thank the authors for their explanations regarding the source, which are satisfactory.*

(R7) We are glad that Reviewer #1 is satisfied with our explanations of the source.

(C8) *I still find that there is a very weak link to negative refraction. As the authors present with the new figures in the supporting material, the effect of negative refraction is seen after propagation through a medium as propagation at negative angles with respect to the normal instead of positive as in conventional refractive media. The current work presents a rotating power flow with opposite orientation with respect to the chiral source exciting the mode. This could be loosely seen as an analogy, but the present work does not study the counter-propagating energy flow outside of the medium, so the analogy is not complete. More importantly, it is not clear what is the usefulness of this analogy.*

(R8) The novel chiral response is a bulk response to a chiral source, rather than a property related to an interface to outside medium. The difference in response to the chiral source in the expanded and shrunken structures is already demonstrated very clearly, in both theory and experiment, as documented in our manuscript. The analogy between it to negative refraction is not crucial for the understanding of the novel chiral response under concern of the present work. Therefore, in our humble opinion, study on geometry including interface between the two structures can be taken as an interesting future problem. As addressed at the very beginning of this rebuttal report (see R0),

the novel chiral responses in the honeycomb structures can potentially be explored for various advanced optical applications. However, whether the analogy itself yields useful insights is not important for the present work.

Finally, we wish to express our heartfelt thanks to Reviewer #1 for spending valuable time to review our work, and raise valuable comments. The course of addressing the criticisms indeed helps us sharpen our presentation. With all the comments addressed, we sincerely hope that our manuscript is ready for publication in Nature Communications.

Rebuttal to the second report of Reviewer #2

The revised manuscript well addresses my concerns. I like the newly added simulation with a large enough structure, which is quite convincing. I'd like to thank the author for correcting my confusion on the Z2 topology. My claim about changing the topology by redefining the unit cell is incorrect. Based on the revision that has been made, I'm glad to support the publishment of the current manuscript.

We wish to express our heartfelt thanks to Reviewer #2 for accepting our replies and revisions in responding to the comments, especially the additional simulations. The support of Reviewer #2 for the publication of our manuscript in Nature Communications is very much appreciated.

REVIEWERS' COMMENTS

Reviewer #1 (Remarks to the Author):

I thank the authors for their replies to my concern. There is still one disagreement point which is the statement that the system can be decomposed in two pseudospins potentially in the whole band. [Phys. Rev. Res. vol. 3, L022013 (2021), Ref. 35, is cited to support this argument, although there are some problems with this paper, as it does not show a mathematical proof that the pseudospins can actually be separated away from Gamma. In any case the authors show the pseudospin character of the modes at Gamma using perturbation theory, and this is enough proof for the physics at play in the current manuscript. As the authors well point out there is no need to consider the Berry curvature along the whole Brillouin zone, unlike for topological properties. Hence I will not oppose to publication as is.

Rebuttal to the third report of Reviewer #1

(C0) *I thank the authors for their replies to my concern.*

(R0) We appreciate the time of Reviewer #1 spent for reviewing our rebuttal and revised manuscript. While our work is recommended for publication by Reviewer #1 as is, we respond the comments in a point-to-point way.

(C1) *There is still one disagreement point which is the statement that the system can be decomposed in two pseudospins potentially in the whole band. [Phys. Rev. Res. vol. 3, L022013 (2021), Ref. 35, is cited to support this argument, although there are some problems with this paper, as it does not show a mathematical proof that the pseudospins can actually be separated away from Gamma.*

(R1) We thank Reviewer #1 for retouching this point. As addressed in the work by Palmer and Gianni [Phys. Rev. Res. vol. 3, L022013 (2021), Ref. 34 in the latest version of our manuscript], pseudospin-up, -down and pseudospinless subspaces are separated in the whole Brillouin zone for the honeycomb photonic crystal of dielectric cylinders proposed in the work by Wu and Hu [Phys. Rev. Lett. vol. 114, 223901 (2015), Ref. 21 in the latest version of our manuscript]. This property is guaranteed mathematically by the combined C_2T symmetry as given explicitly in Eq. (5) in Ref. 34, a symmetry shared by the system under concern in the present work and the dielectric photonic crystal.

(C2) *In any case the authors show the pseudospin character of the modes at Gamma using perturbation theory, and this is enough proof for the physics at play in the current*

manuscript. As the authors well point out there is no need to consider the Berry curvature along the whole Brillouin zone, unlike for topological properties. Hence I will not oppose to publication as is.

(R2) We appreciate Reviewer #1 for accepting our newly added second-order perturbation theory in the last version of manuscript, and our discussion that, unlike topological properties at frequency within the band gap, the phenomena addressed in the present work are predominated by the band properties close to the Gamma point rather than over the whole Brillouin zone. Once again we thank Reviewer #1 very much for recommending publication of our work as is.